# Homologous Recombination-Enhancing Factors Identified by Comparative Transcriptomic Analyses of Pluripotent Stem Cell of Human and Common Marmoset

**DOI:** 10.3390/cells11030360

**Published:** 2022-01-21

**Authors:** Sho Yoshimatsu, Mayutaka Nakajima, Emi Qian, Tsukasa Sanosaka, Tsukika Sato, Hideyuki Okano

**Affiliations:** 1Department of Physiology, Keio University School of Medicine, Tokyo 160-8582, Japan; yoshima@a7.keio.jp (S.Y.); 81710135@keio.jp (M.N.); emisen@keio.jp (E.Q.); sanosaka@keio.jp (T.S.); 81710029@keio.jp (T.S.); 2Laboratory for Marmoset Neural Architecture, RIKEN Center for Brain Science, Saitama 351-0198, Japan

**Keywords:** common marmoset, pluripotent stem cells, homologous recombination

## Abstract

A previous study assessing the efficiency of the genome editing technology CRISPR-Cas9 for knock-in gene targeting in common marmoset (marmoset; Callithrix jacchus) embryonic stem cells (ESCs) unexpectedly identified innately enhanced homologous recombination activity in marmoset ESCs. Here, we compared gene expression in marmoset and human pluripotent stem cells using transcriptomic and quantitative PCR analyses and found that five HR-related genes (BRCA1, BRCA2, RAD51C, RAD51D, and RAD51) were upregulated in marmoset cells. A total of four of these upregulated genes enhanced HR efficiency with CRISPR-Cas9 in human pluripotent stem cells. Thus, the present study provides a novel insight into species-specific mechanisms for the choice of DNA repair pathways.

## 1. Introduction

Repairing DNA double strand breaks (DSBs) is indispensable for maintaining genomic stability [1]. From prokaryotes to eukaryotes, conserved repair pathways for DSBs have been identified: nonhomologous end-joining (NHEJ); homology-directed repair (HDR), including homologous recombination (HR); and microhomology-mediated end-joining (MMEJ), also known as alternative nonhomologous end-joining. The choice of which repair pathways are employed relies on a variety of factors, including DSB complexity, cell type, cell cycle, and species [1,2,3,4].

The genome editing CRISPR-Cas9 technology [5] relies on endogenous pathways for repairing DSBs in cells [6]. Although the combination of CRISPR-Cas9 and double-stranded DNA (dsDNA) donor enables precisely targeted integration or deletion of long sequences by HDR, the efficiency of this process is limited by competing DSB repair pathways, mainly NHEJ. Previous studies have demonstrated that the efficiency of Cas9-meditated HDR can be modified through use of small molecules that stimulate the HDR-related factor RAD51 [7,8] or that inhibit the NHEJ-related factor LIG4 [9,10]. Additionally, modification of Cas9 by fusion with DSB repair-related factors such as RAD52, CtIP, MRE11A [11,12,13], or a dominant negative mutant of P53BP1 [14] can improve HDR efficiency. Moreover, overexpression of several DSB-related genes, including *RAD52* [12] and *RAD18* [15], reportedly contributed to enhancement of Cas9-mediated HDR efficiency.

Human pluripotent stem cells (hPSCs), including human embryonic stem cells (hESCs) and induced pluripotent stem cells (hiPSCs), hold considerable promise for a wide range of applications in the fields of regenerative medicine and stem cell biology [16,17,18,19]. Moreover, knock-in gene targeting (KI) mediated by HR in PSCs offers a powerful approach to the directed differentiation of PSCs [20,21,22,23,24], disease modeling [25,26,27], and gene therapy [28,29,30]. However, although the application of CRISPR-Cas9 greatly facilitates HR-mediated KI in hPSCs, there is still room for further improvement, such as enhancing the ratio of homologous recombinants to total drug-resistant clones, including non-recombinants (genomic integration of a drug resistance cassette by NHEJ-mediated random integration; hereinafter HR ratio). In hPSCs, several studies have shown that even with use of site-specific nucleases, such as zinc finger nucleases (ZFNs), transcription activator-like effector nucleases (TALENs), and CRISPR-Cas9, the HR ratio is less than 50% and generally closer to 30% [31,32,33,34]. Therefore, to achieve robust KI in various loci in hPSCs, there is an urgent need for augmenting the HR ratio; ultimately, complete KI without genotyping confirmation may transform the current status of genome editing technologies.

In a previous study [35], we showed that common marmoset (marmoset; *Callithrix jacchus*) ESCs (cmESCs) harbor an unusual facility for DSB repair, which is characterized by a high HR ratio. In KI experiments using a *PLP1-EGFP* vector targeting the 1st exon of the *PLP1* gene, we found rates of homologous recombinants of 90% with the use of CRISPR-Cas9 and 80% without its use [35]. Moreover, we observed high HR ratios in a variety of other loci, including *ACTB*, *PLP1* (targeting the 2nd, 5th, and 6th exons), *FOXP2*, *PRDM1*, *DPPA3* [20,35,36], and *NANOS3* (Appendix A). Based on these results, we have now explored possible mechanisms for the HR-biased DSB repair in cmESCs. Through use of interspecies analyses of gene expression, we have identified factors that enhance the HR ratio with CRISPR-Cas9 in hiPSCs by ectopic overexpression.

## 2. Materials and Methods

### 2.1. Ethical Statement

Recombinant DNA experiments were approved by the Recombinant DNA Experiment Safety Committee of Keio University (approval number: 27-023 and 27-034).

### 2.2. Vector Construction

The human *hypoxanthine-guanine phosphoribosyltransferase* (*HPRT*) targeting vector HPRT-TV was constructed using the GKI-2.0 system [20]. In brief, based on information from the NCBI genome browser (GRCh38.p13, Chromosome X—NC_000023.11: 134460165...134500668), the 5’ homology arm (3.0 kb) and 3’ homology arm (12.8 kb), which are around the 2nd intron of the gene, were subcloned into pDONR-P3P1r (Addgene #141014) and pDONR-P2rP4 (Addgene #141015), respectively, using the Gateway cloning system (BP reaction). The targeting vector containing a *PGK-Neo* drug-resistance cassette flanked by the *HPRT* homology arms was constructed by Gateway LR reaction using pENTR-L1-PGK-Neo-L2 (Addgene #141007), pUC-DEST-R3R4^®^ (Addgene #141010), and the Gateway LR Clonase II Enzyme Mix (Thermo Fisher Scientific, Waltham, MA, USA). In addition, a human *PROX1-Venus* targeting vector (PROX1-TV) was modified from the *PROX1-tdTomato* vector described in a previous study [20]. The Cas9/sgRNA vector for human *PROX1* locus was also described in the previous study.

A marmoset *NANOS3-Venus* targeting vector was constructed using the GKI-3.0 system [20], which contained the monomeric *Venus* (an improved version of *yellow fluorescent protein*) [37,38] encoding gene and a puromycin/thymidine kinase positive/negative selection cassette flanked by homology arms of the marmoset *NANOS3* gene. In brief, based on information from the NCBI genome browser (GCF_000004665.1, Chromosome 22—NC_013917.1: 13114410…13117680), the 5’ homology arm (540 bp) and 3’ homology arm (542 bp) were cloned into pENTR2-L3-SfoI-Venus-PBL-R1 (Addgene #141019) and pDONR-P2rP4 (Addgene #141015), respectively, using the Seamless Cloning method (Thermo Fisher Scientific). The targeting vector containing a *PGK-PuroTK* cassette flanked by the corresponding homology arms was constructed by Gateway LR reaction using pENTR-L1-PGK-PuroTK-L2 (Addgene #141005) and pUC-DEST-R3R4(R) (Addgene #141010), the Gateway LR Clonase II Enzyme Mix.

PX459 was used as a Cas9/gRNA vector [39]; the vector contained either the human *HPRT* sgRNA sequence TGTTTCAATGAGAGCATTAC or the marmoset *NANOS3* sgRNA sequence TCCTTCCATGTCCACCTAGG.

*RAD51*, *RAD51C*, and *RAD51D* overexpression vectors were constructed using a pDONR-R4-CAG-pA-L2 backbone (kindly provided by Takefumi Sone at Keio University). *BRCA1* and *BRCA2* overexpression vectors were constructed using pcBRCA1-385 (a gift from Lawrence Brody, Addgene plasmid #61586) and pcDNA3 236HSC WT (BRCA2) (a gift from Mien-Chie Hung, Addgene plasmid #16246) [40], respectively, as backbones. SV40 promoter-driven neomycin-resistance cassettes in pcBRCA1-385 and pcDNA3 236HSC WT (BRCA2) were truncated and destroyed by *Sma*I and *Sfo*I digestion and subsequent ligation.

### 2.3. Cell Culture, Transfection, and Drug Selection

A total of three cmESC lines were used: No. 40 (CMES40) and No. 20 (CMES20) [41], and DSY127 [36]; all three cell lines were cultured as described previously [35]. Vector transfection into cmESCs and puromycin selection were performed as described previously [35].

A total of three human iPSC lines, 201B7 [42], WD39 [43] and etKA4 [44], and a human ESC line KhES-1 [45] were used; the cell lines were cultured as described previously [44]. In brief, PSCs were cultured on mitomycin-C-treated G418-resistant SNL76/7 feeder cells [46] in ESM under 20% O_2_ and 5% CO_2_ at 37 °C. ESM consisted of 1× DMEM/F12 (Thermo Fisher Scientific) supplemented with 20% Knockout Serum Replacement (Thermo Fisher Scientific), 0.1 mM MEM Non-Essential Amino Acids Solution (Nacalai Tesque, Tokyo, Japan), 1 mM L-glutamine (L-glu; Nacalai Tesque), 0.1 mM 2-mercaptoethanol (2-ME; Sigma, St. Louis, MO, USA), 100 U/mL penicillin and 100 µg/mL streptomycin sulfate (Nacalai Tesque), and 4 ng/mL basic fibroblast growth factor (Peprotech).

For passaging, cells were pre-treated with 10 μM Rho-associated coiled-coil-containing protein kinase inhibitor Y-27632 (Wako, Tokyo, Japan) in ESM at 37 °C for an hour. The cells were then incubated in CTK solution (Reprocell) at 37 °C for 30 s, mechanically separated from feeder cells, and dissociated by gentle pipetting. The isolated cells were plated onto new feeder cells in ESM supplemented with 10 μM Y-27632. After twenty-four hours, Y-27632 was removed from the medium. Medium change was performed daily. Prior to the seeding of hESCs and iPSCs, feeder cells were seeded onto a gelatin-coated 100 mm cell culture dish in Dulbecco’s modified Eagle’s medium (Thermo Fisher Scientific) supplemented with 10% inactivated fetal bovine serum (Thermo Fisher Scientific).

For transfection (Day 0), we used the NEPA21 Super Electroporator (Nepagene) as described previously [20]. A total of ten μg of HPRT-TV and 5 μg each of the Cas9/gRNA and overexpression vectors were diluted in 100 μl of OPTI-MEM (Thermo Fisher Scientific). The etKA4 hiPSCs (>1 × 10^7^ cells) were suspended in the solution and subjected to electroporation. Transfected cells were plated onto new mitomycin-C-treated G418-resistant SNL76/7 feeder cells on a 100-mm cell culture dish in ESM supplemented with 10 μM Y-27632. On Day 2, selection was initiated by adding 100 ng/mL G418 (an analog of neomycin; Sigma) to the ESM; selection was performed for six days. On Day 8, the concentration of G418 was doubled, and selection was continued for an additional three days. On Day 11, the number of G418-resistant colonies was counted. The cells were then subjected to further selection in 10 μM 6-thioguanine (6TG) for five days. On Day 16, the number of 6TG-resistant colonies was counted. Experimental data were only included when at least 80 G418-resistant colonies survived.

For cell cycle analysis, we used the Fucci2.1 system [47] by lentiviral transfection of CSII-EF-mCherry-hCdt1 and CSII-EF-AmCyan-hGem to the etKA4 iPSCs, and one double-positive clone was chosen and used for further transfection of no-DNA (termed as “sham”) or the BRCA2 vector as described above. We used a FACSVerse (BD Biosciences, New Jersey, United States) for the cell cycle analysis, with 7-AAD (Thermo Fisher Scientific) for removing dead cells.

### 2.4. Genotyping

Southern blotting was performed as described previously [35] using the digoxigenin (DIG) probe system. Genomic DNA samples were digested with *Bgl*II and *Eco*RV by overnight 37 °C incubation. We used the PCR DIG Probe Synthesis Kit (Roche) for DIG-labeled probe production. The human *HPRT*-specific probe (492 bp) was amplified from human genomic DNA using the primers TGCATATCTGGGATGAACTCTGG and AAATGGGACATTTGTGTGTCACC. Molecular sizes were confirmed using DIG-labeled DNA Molecular Weight Marker II (λDNA with *Hin*d III digestion) (Sigma; #11218590910). Genotyping PCR and DNA sequencing analysis (Appendix A) were performed as described previously [20].

### 2.5. Quantitative Reverse-Transcription PCR (qPCR)

RNA extraction, reverse transcription, and PCR were performed as described previously [35]. A total of three biological and technical repetitions of the qPCR analyses were performed. Quantification was performed using the relative standard curve method and endogenous expression of *glyceraldehyde 3-phosphate dehydrogenase* (*GAPDH*) was used as an internal control. Primers were newly or previously designed to amplify both human and marmoset cDNA sequences; however, they did not amplify murine cDNA sequences to avoid contamination due to the use of MEFs for PSC culture [35,48]. The following primers were used: *GAPDH*-forward (GCACCGTCAAGGCTGAGAAC), *GAPDH*-reverse (TGGTGAAGACGCCAGTGGA), *RAD51*-forward (GTCACCTGCCAGCTTCCCATT), *RAD51*-reverse (AGCAGCCGTTCTGGCCTAAAG), *RAD51C*-forward (CGCTGTCGTGACTACACAGAGT), *RAD51C*-reverse (AGGCTGATCATTTGCTGGGCT), *RAD51D*-forward (GGTGCTGCTGGCTCAGTTCT), *RAD51D*-reverse (CGCTACCTGGGCCTCCTACA), *BRCA1*-forward (ACCCGAGAGTGGGTGTTGGA), *BRCA1*-reverse (GCTGTGGGGGATCTGGGGTA), *BRCA2*-forward (TGGGCTCTCCTGATGCCTGTA), and *BRCA2*-reverse (GTATACCAGCGAGCAGGCCG).

### 2.6. RNA-Seq Analysis

Transcriptome data were obtained from cmESCs as described previously (GSE138944) [49]. We used the deposited RNA-seq data from human ESCs and iPSCs (GSE53096) [50] as a reference. Marmoset mRNA was sequenced on an Illumina HiSeq2500 and the obtained nucleotide sequences were mapped against the *Callithrix jacchus* genome (Callithrix_jacchus_cj1700_1.1; https://www.ncbi.nlm.nih.gov/assembly/GCF_009663435.1/ (accessed on 1 December 2021)) by *STAR* (ver.2.5.3a). The number of mapped reads was counted by *featureCounts* (1.5.2) and simultaneously normalized by the TMM method in the *edgeR* package in *R* [51]. The normalized expression levels processed to log2 and z-scoring were visualized using the pheatmap library. In the statistical analysis, Welch’s *t*-test was performed between normalized gene expression levels of human samples and marmoset samples, and the resulting *p* values were processed with Bonferroni correction to obtain the adjusted *p* value. In the present study, adjusted *p* values less than 0.05 were defined as significant; adjusted *p* values processed to −log10 were visualized on the vertical axis by the *ggplot2* library in *R*.

### 2.7. Western Blotting

Western blotting was performed using the Wes—Automated Western Blots with Simple Western (ProteinSimple) according to the manufacturer’s introductions. As primary antibodies, we used polyclonal Rad51 H-92 antibody (1:50 dilution; sc-8349; Santa Cruz) and monoclonal α-tubulin antibody (1:25000 dilution; T9026; Sigma), which was used as an internal control for RAD51 protein quantification. ImageJ software was used to quantify the intensities of Rad51 (37 kDa) and α-tubulin (50 kDa) bands, and then RAD51 expression was normalized against α-tubulin expression.

### 2.8. Statistical Analysis

All data in this study are expressed as means ± S.D. Statistically significant differences were determined using Welch’s *t*-test; *p* values < 0.05 are designated by *; *p* values <0.01 are designated by **, and are interpreted as statistically significant.

## 3. Results

### 3.1. Comparative Transcriptomic Analysis of Human and Marmoset PSCs

Previously, we reported that cmESCs have an innately high HR activity [35]. In particular, extraordinarily high HR ratios for the 1st exon of *PLP1* in a targeting experiment (92.3% with CRISPR-Cas9 against 88.6% without its use) were observed. We also observed high HR ratios using CRISPR-Cas9 in other loci, such as *ACTB*, *PLP1* (targeting the 2nd, 5th, and 6th exons), *FOXP2*, *PRDM1*, *DPPA3*, and *NANOS3* [20,35,36] (see Appendix A). Here, we investigated possible factors that might underlie this phenomenon.

Initially, we investigated HR- and NHEJ-related gene expression in human and marmoset PSCs. In hPSCs, several studies have shown that the HR ratio is less than 50%, generally around 30%, even with use of site-specific nucleases such as ZFN, TALEN, and CRISPR-Cas9 [31,32,33,34]. We used RNA-seq data of hESCs iPSCs deposited in databases [50] to compare gene expression with cmESCs as described previously [49]. We merged and normalized PSC RNA-seq data derived from both species.

HR- and NHEJ-related genes that can be used for comparing fold changes between hPSCs and cmESCs have been listed by hsa03440 and ko03450 in KEGG (https://www.genome.jp/kegg/pathway.html (accessed on 1 December 2021)). Some related genes are absent from the gene lists due to the incomplete gene assembly of the latest version of the marmoset genome (Callithrix_jacchus_cj1700_1.1). Here, we analyzed 37 HR-related and 12 NHEJ-related genes (Figure 1A–D, Appendix A). As summarized in Appendix A, eleven genes (*RAD51D*, *BRCA1*, *BRCA2*, *BABAM1*, *RAD51*, *RAD51C*, *POLD2*, *RAD51B*, *MUS81*, *POLD1,* and *XRCC3*) were significantly up-regulated in cmESCs (Figure 1A), whereas eleven others (*RPA2*, *ATM*, *XRCC2*, *SSBP1*, *RPA3*, *PALB2*, *BRIP1*, *NBN*, *RAD54B*, *UMC1,* and *BRCC3*) were down-regulated (Figure 1B). With regard to NHEJ-related genes, six genes (*XRCC6*, *PRKDC*, *POLM*, *XRCC5*, *RAD50,* and *DNTT*) were significantly up-regulated in cmESCs (Figure 1C), and five others (*FEN1*, *DOLRE1C*, *NHEJ1*, *XRCC4,* and *POLL*) were down-regulated (Figure 1D). In light of the high HR activity in cmESCs, we then focused on HR-related genes that showed increased expression in cmESCs compared to hPSCs. To validate the results of the transcriptome analysis, we performed an interspecies qPCR analysis using primer sets specifically designed for these human and marmoset genes. We also designed primers specific for human and marmoset *GAPDH* for normalization. Preliminary screens using the designed primers showed that an interspecies comparison of expressions was not feasible for several genes (*BABAM1* and *POLD1/2*) owing to a lack of accuracy based on the post-qPCR melt curve analysis (data not shown).

By qPCR using total RNAs from three cmESC lines (No. 40, No. 20, and DSY127) and four hPSC lines (201B7, WD39, KhES-1, and etKA4), we confirmed the significantly higher expression of *RAD51D*, *BRCA1*, *BRCA2*, *RAD51C*, and *RAD51* in cmESCs compared to hPSCs (Figure 2). In particular, *RAD51C* and *RAD51D* expression in cmESCs was approximately 10 and 7 times higher, respectively, than those of hPSCs (Figure 2, top). Quantitative Western blotting confirmed the high RAD51 expression in cmESCs at the protein level (Appendix A).

### 3.2. Enhancement of the HR and RI Ratio with CRISPR-Cas9 in hiPSCs by Overexpression of the Defined Factors

To explore the effect of high expression of the five HR-related genes (*RAD51D*, *RAD51C*, *BRCA2*, *RAD51*, and *BRCA1*) in cmESCs, we induced overexpression of the genes in a gene targeting experiment with *HPRT* in the male hiPSC line etKA4 [44]. The HPRT protein catalyzes the salvage pathway, synthesizing inosine monophosphate and guanosine monophosphate from hypoxanthine and guanine, respectively [52]. HPRT deficiency results in the loss of susceptibility for 6TG, a toxic analog of guanine [53,54]. We selected the HPRT targeting system as it has been frequently used to assess the HR ratio in mammalian male PSCs [24,55,56].

We constructed a knock-in/knock-out system for the human *HPRT* gene, which is located on the X chromosome (Figure 3A,B). The targeting vector (HPRT-TV) harbored a neomycin-resistance (PGK-Neo) cassette flanked by 3.0 kb 5’ homology and 12.8 kb 3’ homology arms (Figure 3A). Following KI, the 2nd exon of *HPRT* was completely replaced with a PGK-Neo cassette, which resulted in the loss of functional mRNA expression from the *HPRT*^Neo^ allele (Figure 3A). Initial G418 selection (both homologous recombinants and non-recombinants survived) and subsequent 6TG selection (only homologous recombinants survived) enabled robust quantification of the HR ratio without the necessity of genotyping individual clones (Figure 3B). In addition, we constructed a PX459 (Addgene #62988)-based Cas9/gRNA vector [39] containing the sgRNA sequence for the 2nd intron of *HPRT*, which did not recognize HPRT-TV.

As it is possible that the genomic cleavage of the *HPRT* 2nd intron after transfection of the Cas9/gRNA vector and subsequent NHEJ or MMEJ-mediated introduction of small and large deletion could produce an undesired knock-out allele, we initially tested transfection of only the Cas9/gRNA vector into the etKA4 hiPSCs. In this initial test, no 6TG-resistant colonies were obtained from 1 × 10^7^ transfected cells (*n* = 3), showing that the NHEJ or MMEJ-mediated deletion in the intronic region has a negligible effect with regard to the assessment of the HR ratio in the hiPSCs.

After co-transfection of the Cas9/gRNA vector and HPRT-TV, and serial G418 and 6TG selection, we confirmed that all analyzed G418 and 6TG-resistant (NeoR+6TGR) clones were hemizygous *HPRT*^Neo^ recombinants by Southern blotting (Figure 3C).

Because we wished to evaluate the effects of single and multiple overexpression of the five genes on HR ratios (Figure 3B), we used a CAG-EGFP vector [49] as a mock control for the overexpression vectors. Initially, we transfected HPRT-TV and each overexpression vector (or mock) and quantified HR ratios without CRISPR-Cas9. None of the attempts at single and multiple overexpression of the five genes enhanced HR ratios significantly (Figure 3D; mock, 0.0033 ± 0.0018; +RAD51, 0.0060 ± 0.0040; +BRCA1, 0.0060 ± 0.0030; +BRCA2, 0.0026 ± 0.0043; +RAD51C, 0.0052 ± 0.0052; +RAD51D, 0.0048 ± 0.0035; +RAD51C/D, 0.0045 ± 0.0029; +BRCA1/2&RAD51C/D, 0.023 ± 0.023; *n* ≧ 4).

Next, we quantified HR ratios with use of CRISPR-Cas9. When the mock vector was transfected with HRPT-TV and the Cas9/gRNA vector, an HR ratio of approximately 30% was achieved (Figure 3D; 0.336 ± 0.026, *n* = 4); this HR ratio is comparable with previous results using hPSCs [31]. Overexpression of only *RAD51* did not result in a significant enhancement of the HR ratio (0.256 ± 0.163, *n* = 3).

In comparison to use of the mock, single overexpression of *BRCA1* or *BRCA2* surprisingly resulted in a significant enhancement of the HR ratio (Figure 3D; +BRCA1, 0.590 ± 0.036; +BRCA2, 0.582 ± 0.034; *n* = 3). In addition, co-overexpression of *BRCA1* and *BRCA2* also enhanced the HR ratio (0.563 ± 0.049, *n* = 3), while co-overexpression of *BRCA1/2* and *RAD51* did not produce an enhanced HR ratio (0.265 ± 0.049, *n* = 3).

Single overexpression of *RAD51C* or *RAD51D* did not result in a significant enhancement of the HR ratio (Figure 3D; +RAD51C, 0.468 ± 0.051; +RAD51D, 0.414 ± 0.073; *n* = 4). However, when the two genes were co-overexpressed, we observed an enhanced HR ratio (0.560 ± 0.065, *n* = 3); however, this effect was not observed when *RAD51* was also co-overexpressed (0.185 ± 0.03, *n* = 3).

We tested multiple sets of overexpression of the five genes (Figure 3D). Except when RAD51 was co-transfected with the other four genes (0.388 ± 0.032, *n* = 3), co-overexpression of *RAD51C/D* and *BRCA1/2* resulted in significant enhancement of HR ratios: +RAD51C and BRCA1, 0.681 ± 0.062; +RAD51C and BRCA2, 0.706 ± 0.080; +RAD51D and BRCA1, 0.607 ± 0.048; +RAD51D and BRCA2, 0.529 ± 0.006; +RAD51C and BRCA1/2, 0.516 ± 0.021; +RAD51D and BRCA1/2, 0.540 ± 0.024; +RAD51C/D and BRCA1, 0.498 ± 0.048; +RAD51C/D and BRCA2, 0.539 ± 0.044 (*n* = 3 for all combinations). Finally, we demonstrated that co-overexpression of the factors *BRCA1*, *BRCA2*, *RAD51C*, and *RAD51D* resulted in a significant enhancement of the HR ratio (0.686 ± 0.061, *n* = 3).

Last, we performed additional KI experiments in another locus, *PROX1*. We used the Cas9/gRNA vector and *PROX1-Venus* targeting vector (PROX1-TV), which was slightly modified from a previous study [20]. By the construct, *2A-Venus* and a puromycin-resistance cassette are introduced into the termination codon (exon 5) of the *PROX1* gene (Appendix A). We transfected the Cas9/gRNA and PROX1-TV with or without the overexpression vectors of five (*RAD51*, *RAD51C*, *RAD51D*, *BRCA1,* and *BRCA2*) factors. Following puromycin selection, hiPSC colonies, putative KI, or WT clones were genotyped by PCR (Appendix A). The results of the experiments are summarized in Appendix A—3/12 for control (no overexpression), 5/12 for *RAD51* overexpression, 6/12 for *RAD51C* overexpression, 7/12 for *RAD51D* or *BRCA2* overexpression, 8/12 for *BRCA1,* or four factor (*RAD51C/D* and *BRCA1/2*) overexpression. Again, the overexpression of four factors enhanced KI efficiency in hiPSCs, which shows the robust applicability of the technology for further applications. The precise integration of *2A-Venus* was confirmed by Sanger Sequencing (Appendix A). The precise knock-in at the *HPRT* locus (described in Figure 3) was also confirmed by genotyping PCR and Sanger sequencing (Appendix A). We also note that the overexpression of *BRCA2* may have directly or indirectly affected the perturbation of hiPSCs’ cell cycle (Appendix A).

## 4. Discussion

In this study, through comparative analyses of gene expression in human and marmoset PSCs, we have identified four genes (RAD51C, RAD51D, BRCA1, and BRCA2) whose single and multiple overexpression increased HR ratios in hiPSCs. Intriguingly, we also observed that overexpression of RAD51 did not enhance the HR ratio in hiPSCs; an alternative explanation is that RAD51 overexpression may cancel the effect of the enhancement of the HR ratio by the other four genes. In the Appendix A, we further discuss how these genes involve in the HR machinery, and the discrepancy of the effect of RAD51 overexpression from the part of previous studies. Results presented here also suggest the possibility of a vice versa effect. In fact, we demonstrated the overexpression of four factors (*RAD51C*, *RAD51D*, *BRCA1*, and *BRCA2*), which were highly expressed in cmESCs, and which contributed to the enhancement of HR ratios in hPSCs. Thus, it is also possible that overexpression of several NHEJ factors, which were lowly expressed in cmESCs (including *FEN1*, *DCLRE1C* (*Artemis*), *NHEJ1*, and *XRCC4*), may contribute to decreased HR ratios in hPSCs. Further analyses are required to evaluate the robust effects of the four HR factors, such as effects on KI in other loci, and in different cell lines and species; nevertheless, our investigation has demonstrated that overexpression of these factors may ameliorate the HR ratio with CRISPR-Cas9 in hiPSCs. We also note that, in the present study, we could not scrutinize the risk of HR factor overexpression for the usage of iPSCs in other experimental settings, including transplantation and drug-screening analysis.

In clinical settings, although knock-in technology in donor PSCs is beneficial due to the highly customizability, constitutive overexpression of exogenous gene(s) may impose potential risks, including tumorigenicity and genome instability. In this context, the critical time window of HR-factor overexpression for increasing HR ratios should be assessed in further studies.

## Figures and Tables

**Figure 1 cells-11-00360-f001:**
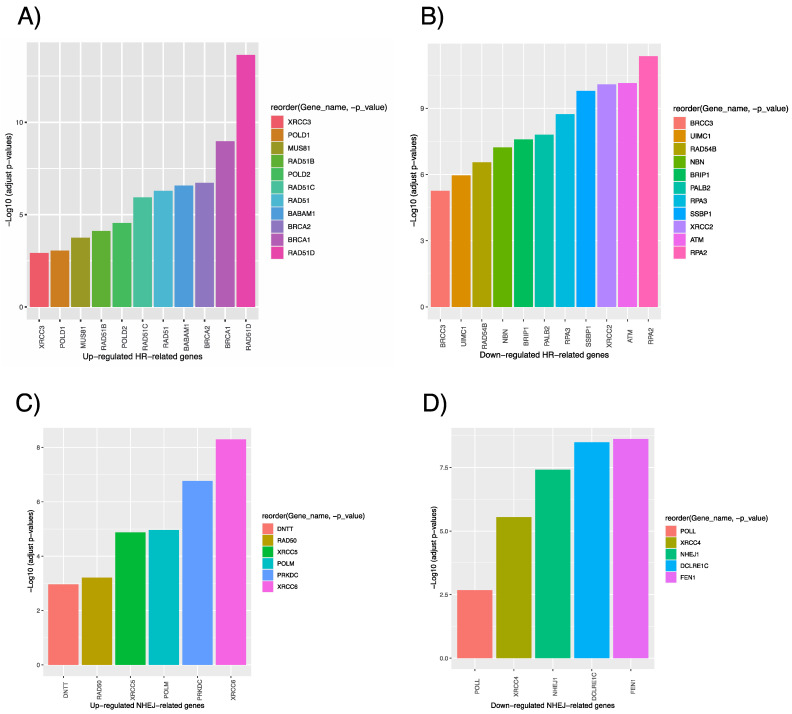
Differential expression of DSB repair genes in human and marmoset PSCs. (**A**,**B**) HR-related genes that are significantly upregulated or downregulated in cmESCs compared to those in hESCs and iPSCs. Gene groups are referenced to KEGG hsa03440. *Y*-axis shows adjusted *p*-values after −log10 treatment. (**C**,**D**) NHEJ-related genes showing significantly higher or lower expression levels in cmESCs compared to hESCs and iPSCs. Gene groups refer to KEGG ko03450. *Y*-axis shows the adjusted *p*-value after −log10 treatment.

**Figure 2 cells-11-00360-f002:**
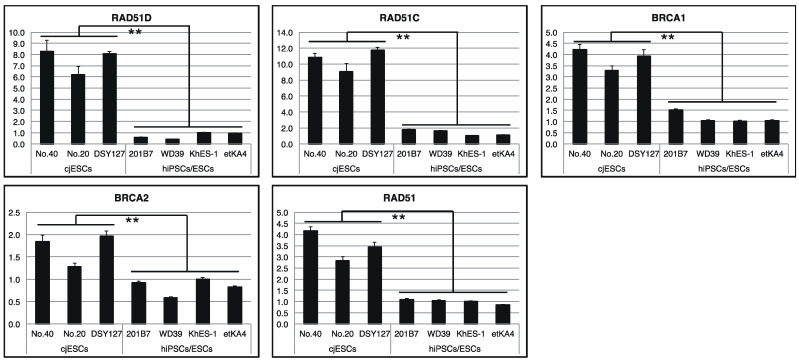
Interspecies qPCR analysis for *RAD51C*, *RAD51D*, *BRCA1*, *BRCA2,* and *RAD51*. RQ values of each human and marmoset sample (biological and technical triplicates) were used for the statistical significance tests (**, *p* value < 0.01).

**Figure 3 cells-11-00360-f003:**
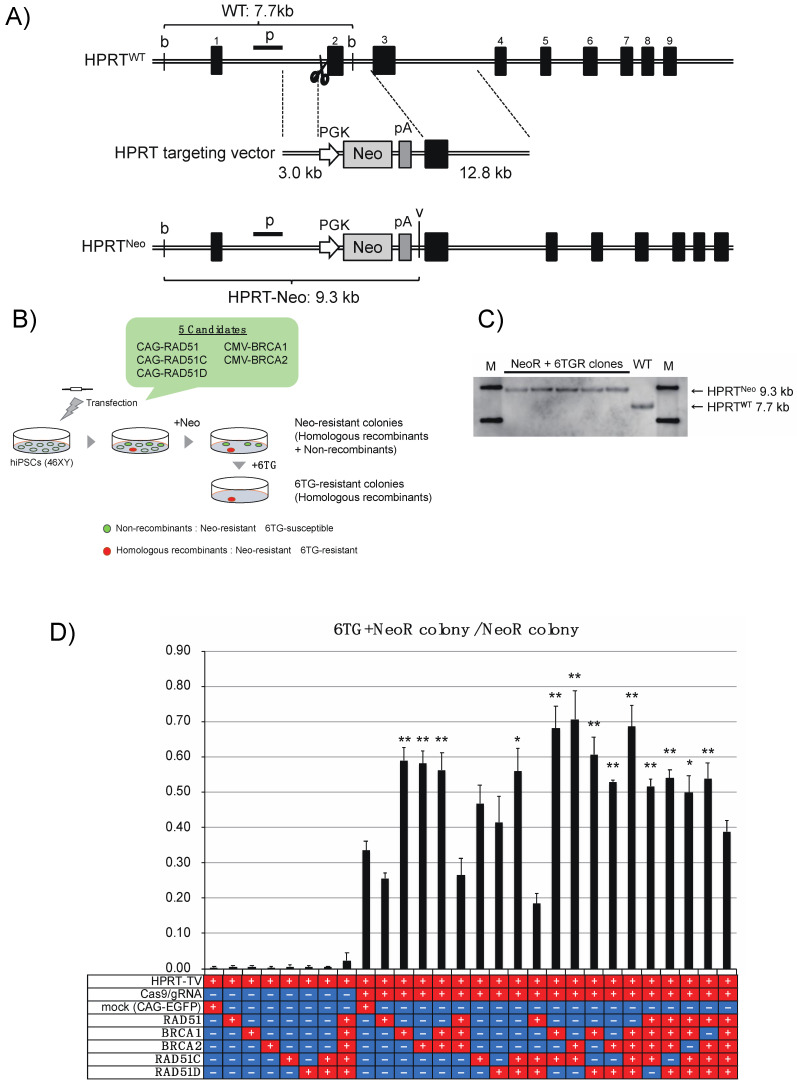
*HPRT* targeting in hiPSCs. (**A**) Graphical schematics of the wild-type human *HPRT* locus (*HPRT*^WT^; **top**), *HPRT* targeting vector (**middle**), and recombinant *HPRT* locus (*HPRT*^Neo^; **bottom**). Black boxes indicate endogenous exons of the *HPRT* gene (upper numbers indicate each exon number). *PGK*, mouse *phosphoglycerate kinase 1* promoter; *Neo*, *neomycin resistance gene*; *pA*, *polyadenylation signal sequence*; b, *Bgl*II recognition site; v, *Eco*RV recognition site; p, the probe site for Southern blotting analysis. (**B**) Graphical schematic of the HPRT targeting experiment using hiPSCs. *CAG*, *CAG promoter*; *CMV*, *human cytomegalovirus immediate early enhancer and promoter*. (**C**) Southern blotting analysis of genomic DNA derived from G418/6TG-double-resistant five clones and wild-type (WT) hiPSCs. M, DNA marker (λDNA with *Hin*d III digestion). (**D**) Resultant HR and RI ratios in the HPRT targeting experiments. Values were calculated as 6TG+NeoR colony number and NeoR colony number. Asterisks indicate statistical significance in comparisons of Cas9/gRNA (+) samples of control versus Cas9/gRNA (+) mock (+). Asterisks show the statistical significance of Cas9/gRNA(+) samples compared to the mock; *, *p* value < 0.05; **, *p* value < 0.01.

## Data Availability

No new data were created or analyzed in this study. Data sharing is not applicable to this article.

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
