# Peer review of "Homologous Recombination-Enhancing Factors Identified by Comparative Transcriptomic Analyses of Pluripotent Stem Cell of Human and Common Marmoset"

_cells, 2022, doi:10.3390/cells11030360_

Round 1

Reviewer 1 Report

I have previously revised this paper. I have asked for two changes: the first was a request of shortening the Discussion; in the second I asked to discuss a limitation about the possibility to transfer this approach to the bedside.

I feel that both my suggestions have been met and therefore I would accept the paper.

Author Response

Author's Reply to the Review Report (Reviewer 1)

We greatly appreciate the Reviewers for providing us constructive comments for our manuscript cells-1524971.

In our response, page and line numbers (x) in the revised version manuscript are indicated as Px, and Lx-x.

Reviewer comment:

> I have previously revised this paper. I have asked for two changes: the first was a request of shortening the Discussion; in the second I asked to discuss a limitation about the possibility to transfer this approach to the bedside.

I feel that both my suggestions have been met and therefore I would accept the paper.

Reply:

We are grateful for these helpful suggestions for improving the manuscript.

Reviewer 2 Report

Dr. Sho Yoshimatsu and colleagues performed original studies, they investigated molecular factors that involved in phenomenon of extraordinarily high HR ratios in marmoset ESCs and identified four genes (RAD51C, RAD51D, BRCA1, and BRCA2) which single and combinatory overexpression increased HR ratios in human iPSCs. They evidenced with RNA-seq and qPCR that expression of RAD51D, BRCA1, BRCA2, RAD51C, and RAD51 is significantly higher in cmESCs compared to hPSCs. With the used of developed HPRT targeting system and CRISPR-Cas9 method the authors demonstrated that co-overexpression of the factors BRCA1, BRCA2, RAD51C, and RAD51D resulted in a significant enhancement of the HR ratio in human PSCs.

The manuscript is well written and organized, all points are strongly supported by experimental data and statistical analysis. The performed studies have an important value for the fields of iPSC-based cell technology and CRISPR/Cas9 technology and homologous recombination in DNA repair research. There are a few minor suggestions that may be considered. 

1) The title of the article is long; it is better to shorten and clarify it. As an example:  Comparative analyses of gene expression in marmoset and human pluripotent stem cells identify factors enhancing homologous recombination efficiency. 
2) In the abstract - (cmESCs), (PSCs), (qPCR), (HR)  are not necessary to indicate.
3) Manuscript published online on biorxiv https://www.biorxiv.org/content/10.1101/2021.04.05.438539v1.abstract
In the submitted version of the manuscript the figures 1 and 2 are not in correct place and empty (black). To this regard only based on the version of the paper from biorxiv became possible to review the manuscript.
4) The section of supported discussion is better to put together with the main discussion section.

Author Response

Author's Reply to the Review Report (Reviewer 2)

We greatly appreciate the Reviewers for providing us constructive comments for our manuscript cells-1524971.

In our response, page and line numbers (x) in the revised version manuscript are indicated as Px, and Lx-x.

Reviewer comment:

> Dr. Sho Yoshimatsu and colleagues performed original studies, they investigated molecular factors that involved in phenomenon of extraordinarily high HR ratios in marmoset ESCs and identified four genes (RAD51C, RAD51D, BRCA1, and BRCA2) which single and combinatory overexpression increased HR ratios in human iPSCs. They evidenced with RNA-seq and qPCR that expression of RAD51D, BRCA1, BRCA2, RAD51C, and RAD51 is significantly higher in cmESCs compared to hPSCs. With the used of developed HPRT targeting system and CRISPR-Cas9 method the authors demonstrated that co-overexpression of the factors BRCA1, BRCA2, RAD51C, and RAD51D resulted in a significant enhancement of the HR ratio in human PSCs.

> The manuscript is well written and organized, all points are strongly supported by experimental data and statistical analysis. The performed studies have an important value for the fields of iPSC-based cell technology and CRISPR/Cas9 technology and homologous recombination in DNA repair research. There are a few minor suggestions that may be considered.

1) The title of the article is long; it is better to shorten and clarify it. As an example:  Comparative analyses of gene expression in marmoset and human pluripotent stem cells identify factors enhancing homologous recombination efficiency.

2) In the abstract - (cmESCs), (PSCs), (qPCR), (HR) are not necessary to indicate.

3) Manuscript published online on biorxiv https://www.biorxiv.org/content/10.1101/2021.04.05.438539v1.abstract

In the submitted version of the manuscript the figures 1 and 2 are not in correct place and empty (black). To this regard only based on the version of the paper from biorxiv became possible to review the manuscript.

4) The section of supported discussion is better to put together with the main discussion section.

Reply:

(1) According to the Reviewer’s suggestions, we have shortened the manuscript title to “Homologous recombination-enhancing factors identified by comparative transcriptomic analyses of pluripotent stem cell of human and common marmoset”, which may remain the significance and novelty of the present study.

(2) Per the helpful suggestion, we shorten the abstract as following:

A previous study assessing the efficiency of the genome editing technology CRISPR-Cas9 for knock-in gene targeting in common marmoset (marmoset; Callithrix jacchus) embryonic stem cells (ESCs) unexpectedly identified innately enhanced homologous recombination activity in marmoset ESCs. Here, we compared gene expression in marmoset and human pluripotent stem cells using transcriptomic and quantitative PCR analyses and found that five HR-related genes (BRCA1, BRCA2, RAD51C, RAD51D and RAD51) were upregulated in marmoset cells. Four of these upregulated genes enhanced HR efficiency with CRISPR-Cas9 in human pluripotent stem cells. Thus, the present study provides a novel insight into species-specific mechanisms for the choice of DNA repair pathways.

(3) We are very sorry for the inconvenience which may result from a bug in our Word software. We included the original (revised) Fig data on the revised manuscript.

(4) We were wondering if we should put the long discussion in the main text or supplementary, but the Reviewer #1 suggested to put it in the latter. Considering the simpleness and length restriction of the manuscript, we decided to put it in the supplementary.

Reviewer 3 Report

In this study, the authors found that the upregulation of four HR-related genes (BRCA1, BRCA2, RAD51C, RAD51D) in marmoset cells and their overexpression increased HR efficiency with CRISPR-Cas9 in human PSCs. It is a very interesting study which would further promote the application of CRISPR-Cas9 technique in human PSCs.

1. The objective of this study needs to be further emphasized, and the risks of other factors in the KI experiments are not mentioned.

2. Figure3D showed HR ratio was still increased significantly after RAD51 overexpression although one of the four HR-related genes was missed. Therefore, the conclusion of “RAD51 overexpression cancelled the effect of the enhancement of the HR ratio by the other four genes” may not rigorous.

3. The authors used Neo resistant and 6TG resistant colony count to measure KI ratios; however, it may be insufficiency because the screening efficiency could not be complete and it is difficult to ensure every colony come from one cell. In addition, sequencing of the positive colonies is also necessary to confirm that they have the accurate KI sequence.

4. The full names of some abbreviations were not given, such as HPRT line 67, PX459 line 91.

5. The Figures S1-3 were not found in the supplementary files.

Author Response

Author's Reply to the Review Report (Reviewer 3)

We greatly appreciate the Reviewers for providing us constructive comments for our manuscript cells-1524971.

In our response, page and line numbers (x) in the revised version manuscript are indicated as Px, and Lx-x.

> In this study, the authors found that the upregulation of four HR-related genes (BRCA1, BRCA2, RAD51C, RAD51D) in marmoset cells and their overexpression increased HR efficiency with CRISPR-Cas9 in human PSCs. It is a very interesting study which would further promote the application of CRISPR-Cas9 technique in human PSCs.

  1. The objective of this study needs to be further emphasized, and the risks of other factors in the KI experiments are not mentioned.

  1. Figure3D showed HR ratio was still increased significantly after RAD51 overexpression although one of the four HR-related genes was missed. Therefore, the conclusion of “RAD51 overexpression cancelled the effect of the enhancement of the HR ratio by the other four genes” may not rigorous.

  1. The authors used Neo resistant and 6TG resistant colony count to measure KI ratios; however, it may be insufficiency because the screening efficiency could not be complete and it is difficult to ensure every colony come from one cell. In addition, sequencing of the positive colonies is also necessary to confirm that they have the accurate KI sequence.

  1. The full names of some abbreviations were not given, such as HPRT line 67, PX459 line 91.

  1. The Figures S1-3 were not found in the supplementary files.

Reply:

(1) According to the helpful suggestion, we added the following sentence in the Introduction section (P2, L60-63):

Therefore, to achieve robust KI in various loci in hPSCs, there is an urgent need for attenuating HR ratio – ultimately, complete KI without genotyping confirmation may transform the current status of genome editing technologies.

We also add the following sentence in the Discussion section (P11, L366-369):

We also note that, in the present study, we could not scrutinize the risk of HR factor overexpression for the usage of iPSCs in other experimental settings, including transplantation and drug-screening analysis.

(2) According to the suggestion, we changed the following sentence in the Discussion section (P10, L352-355):

Intriguingly, we also observed that overexpression of RAD51 did not enhance the HR ratio in hiPSCs; an alternative explanation is that RAD51 overexpression may cancel the effect of the enhancement of the HR ratio by the other four genes.

(3) We agree with the Reviewer, however, it is very (currently) technically difficult to analyze single cell-derived genotype in this experimental setting, because the current whole genome amplification technologies were mainly customized for comparatively short (~2kb) PCR genotyping, which we seriously faced in a subsequent knock-in marmoset study (Yoshimatsu et al., unpublished). In other words, genotyping such as long PCR or Southern blotting used for genotyping of HPRT KI is not feasible in a single-cell level.

To assess the preciseness of the HPRT KI in G418 and 6-TG-resistant iPSCs, we added Figure S6, which shows the almost precise KI in the HPRT locus.

(4) Thank you very much for suggestion. Accordingly, we corrected the corresponding points in the manuscript.

Round 2

Reviewer 3 Report

The previous comments have been addressed well. 

Author Response

We greatly appreciate the Reviewer for kind attention for our manuscript.